# Aminopeptidase N Inhibitors as Pointers for Overcoming Antitumor Treatment Resistance

**DOI:** 10.3390/ijms23179813

**Published:** 2022-08-29

**Authors:** Oldřich Farsa, Veronika Ballayová, Radka Žáčková, Peter Kollar, Tereza Kauerová, Peter Zubáč

**Affiliations:** 1Department of Chemical Drugs, Faculty of Pharmacy, Masaryk University, Palackého 1946/1, 612 00 Brno, Czech Republic; 2Department of Pharmacology and Toxicology, Faculty of Pharmacy, Masaryk University, Palackého 1946/1, 612 00 Brno, Czech Republic

**Keywords:** aminopeptidase N, acetamidophenones, Schiff bases, semicarbazones, thiosemicarbazones, inhibition of proliferation

## Abstract

Aminopeptidase N (APN), also known as CD13 antigen or membrane alanyl aminopeptidase, belongs to the M1 family of the MA clan of zinc metallopeptidases. In cancer cells, the inhibition of aminopeptidases including APN causes the phenomenon termed the amino acid deprivation response (AADR), a stress response characterized by the upregulation of amino acid transporters and synthetic enzymes and activation of stress-related pathways such as nuclear factor kB (NFkB) and other pro-apoptotic regulators, which leads to cancer cell death by apoptosis. Recently, APN inhibition has been shown to augment DR4-induced tumor cell death and thus overcome resistance to cancer treatment with DR4-ligand TRAIL, which is available as a recombinant soluble form dulanermin. This implies that APN inhibitors could serve as potential weapons for overcoming cancer treatment resistance. In this study, a series of basically substituted acetamidophenones and the semicarbazones and thiosemicarbazones derived from them were prepared, for which APN inhibitory activity was determined. In addition, a selective anti-proliferative activity against cancer cells expressing APN was demonstrated. Our semicarbazones and thiosemicarbazones are the first compounds of these structural types of Schiff bases that were reported to inhibit not only a zinc-dependent aminopeptidase of the M1 family but also a metalloenzyme.

## 1. Introduction

APN is sometimes called “a moonlighting enzyme”. It is a widely expressed ectoenzyme with many functions that do not always depend on its enzymatic activity. The membrane-bound form of APN, which is expressed in the renal and intestinal epithelia, the nervous system, myeloid cells, and fibroblast-like cells, such as synoviocytes, is often referred to as hCD13, whereas the soluble form, which is present in human serum at a concentration of about 4.6 nM, is known as sCD13 [1]. There is a strong correlation between the expression and enzymatic activity of hCD13 and sCD13 and the invasive capacity of numerous tumor cell types. APN also serves as a receptor involved in endocytosis during viral infection such as in the human coronavirus HCoV-229E, among others [2]. As a signaling molecule, it takes part in adhesion, phagocytosis, and angiogenic processes [3]. The plasmatic concentration of sCD13 can be used as a prognostic marker for some cancers such as non-small cell lung cancer (NSCLC) including lung adenocarcinoma [4]. APN is a promising target for anticancer therapy. Newer research suggests that it serves as one of the molecular targets for the anticancer antibiotic actinomycin D and its simplified analogs [5]. Bestatin (3-Amino-2-hydroxy-4-phenylbutanoyl)-L-leucine or ubenimex in INN nomenclature, Figure 1, one of the most known APN inhibitors, was first isolated from the bacteria *Streptomyces olivoreticuli* in 1976. It was used as an anticancer agent for the treatment of lung cancer and acute myeloid leukemia in Japan for several years [6,7]. More recently, it has also been demonstrated to inhibit cell proliferation, migration, and invasion in both renal cell carcinoma and prostate cancer [8,9]. Bestatin has also been shown to be capable of attenuating the acquired resistance of renal cell carcinoma to treatment with sorafenib, which is today a first-line therapy for this cancer [10]. Tosedostat, cyclopentyl (2S)-2-[[(2R)-2-[(1S)-1-hydroxy-2-(hydroxyamino)-2-oxoethyl]-4-methylpentanoyl]amino]-2-phenylacetate, a synthetic dipeptide containing the carbohydroxamic group (Figure 1), is also a known APN inhibitor.

It has undergone more than ten clinical trials of phases 1 or 2 for the treatment of myeloid leukemias and solid tumors. Its anticancer activity is mainly attributed to the inhibition of the cleavage of proteins and peptides by M1 family aminopeptidases including APN. This disrupts normal cellular protein turnover, resulting in both peptide accumulation and a decrease in intracellular free amino acid content, a process that appears to preferentially affect metabolically active cells such as malignant cells. Such an inhibition triggers the phenomenon termed the amino acid deprivation response (AADR), a stress response comprising the upregulation of amino acid transporters and synthetic enzymes and the activation of stress-related pathways such as NFkB and other pro-apoptotic regulators [11]. Death receptor 4 (DR4 or TRAIL-R1), a member of the DR subgroup of the tumor necrosis factor (TNF) receptor superfamily, is overexpressed in various types of tumor cells. DR4 mediates extrinsic apoptotic cascades using binding to TNF-related apoptosis-inducing ligands (TRAIL or Apo2L). Unfortunately, resistance is often observed in the clinical application of TRAIL, which has undergone five clinical trials on its soluble recombinant form dulanermin, and another study for the treatment of peritoneal carcinomatosis continues [12]. In a recent study, bestatin markedly sensitized fibrosarcoma cells previously implanted in athymic nude mice to apoptosis induced by TRAIL [13]. Numerous further APN inhibitors were prepared as potential anti-cancer drugs. Recent progress in this field is summarized in a review article [1]. APN belongs among the zinc metallopeptidases. As far as a mechanism of inhibition is concerned, zinc chelation is frequently mentioned. Many reported inhibitors are attributed to this mechanism of action. Hydroxamic acids with the ureido fragment in their molecules [14,15,16] or without [17] use their carbohydroxamic group as the coordinating moiety for Zn^2+^. Vicinal cycloaliphatic amino ketones, specifically 3-amino-2-tetralone derivatives and analogs, use this complexation as their primary amino group together with the oxygen of the adjacent keto group [18]. Semicarbazones and thiosemicarbazones are known zinc chelators [19,20], although this fact has never been used in the design of metalloenzyme inhibitors. In this article, we describe the design, synthesis, and APN inhibition activity of a series of novel, basically substituted acetamidoacetophenone-semicarbazones and -thiosemicarbazones and their starting ketones, with either the dialkylamine group or a saturated nitrogenous heterocycle as a basic substituent in the acetamido part of the molecule.

## 2. Results

### 2.1. Synthesis of Target Compounds

Our target compounds were synthesized by a four-stage synthetic sequence that is depicted in Figure 1.

The starting 2-,3- or 4-aminoacetophenone **ao**, **am**, **ap** reacted with chloroacetyl chloride to give 2-, 3- or 4-(chloroacetamido)acetophenones **12-1**, **13-1**, **14-1**, which were then subjected to the reaction with individual secondary amines to give basically substituted ketones **12-2-1** to **14-11-1** (in case of dialkyl amines as reagents, 2-, 3- or 4-[2-(dialkylamino)acetamido]acetophenones were prepared). The reaction of such ketones with thiosemicarbazide in ethanol without the presence of any strong acid then led to the appropriate thiosemicarbazones as bases (**22-5-1–24-9-1**), whereas the analogous reaction with semicarbazide gave semicarbazones as bases, **32-2-1–34-10-1**. Analogous reactions with thiosemicarbazide in the presence of either perchloric or hydrochloric acid then led to appropriate thiosemicarbazones in the form of perchlorate or hydrochloride, **23-7-3–24-11-3.** The compounds were purified by a simple crystallization from the system ethanol/water with the addition of charcoal in the case of need and characterized with 1H- and 13-C-NMR, IR, and MS spectra. Two-dimensional NMR spectra (H-H-cosy, HMQC, HMBC) were used for the 1D NMR spectra interpretation. NOESY (NMR) spectra were used to determine the geometry of the Schiff bases. They revealed that the prepared products consisted of about equimolar amounts of *E* and *Z* (or *syn*/*anti*) isomers. The structural characteristics of the prepared compounds can be found in the Appendix A as well as the procedure for the determination of their purity, and the yields and values of the purity of the target compounds and key intermediates are summarized in Table 1.

### 2.2. APN Inhibitory Activity and QSAR in It

#### 2.2.1. Determination of APN Inhibition

APN inhibitory activity was determined using a standardized spectrophotometry protocol using L-leucine-p-nitroanilide as a chromogenic substrate for APN. Measurements were performed in triplicate at 405 nm at a Cytation 3 well-plate reader. The results were processed into IC_50_ values using GraFit 5 software and are listed in Table 2 below.

In some compounds, solubility problems occurred, which complicated the inhibition activity determination. We overcame these either by the use of cosolvents or by an alternative HPLC approach [21]. The details are summarized in the Appendix A.

#### 2.2.2. QSAR in APN Inhibitory Activity

The classical Hansch method of regression analysis was used to determine the dependence of the inhibitory expressed as IC_50_ on the important structure descriptors. Typically, the activity of the members of a homologous series of biologically active compounds correlates with their lipophilicity. Furthermore, in our case, a hint of such a correlation was also found. This situation was expressed by Equation (1):log IC_50_ = 0.6925 log P + 0.5684(1)
where log P is calculated by an algorithm based on >12,000 experimental logP values using the principle of isolating carbons [22], as a parameter of lipophilicity was used. The low determination coefficient and F-statistic values (R^2^ = 0.4033, F = 3.2615) indicated that the correlation was insufficient and a further structure parameter had to be added. An electronic parameter was then used because it was very probable that dissociation at various sites of the molecule could have an impact on coordination to Zn^2+^ cation as well as on interactions with the acidic or basic parts of the amino acid residues of the enzyme protein. Since our target compounds have two centers of acidity and at least two centers of basicity, it was more advantageous to express the electronic properties of a molecule with one electronic parameter, the isoelectric point pI, than with a set of dissociation constants. The set of pI values computed by the algorithm implemented in the Marvin software [23] was chosen for such a purpose. Ketones **14-2-1**, **14-6-1,** and **14-8-1** were preliminarily excluded because there was an assumption of a different Zn^2+^ complexation in them than in the thiosemicarbazones and semicarbazones. Further, **24-4-1** and **34-8-1** were excluded as outliers during the regression analysis. Finally, a regression model with both parameters squared was found. It is expressed by Equation (2):logIC_50_ = −0.4475(logP)^2^ − 0.1452(pI)^2^ + 1.8847(logP)(pI) − 0.6101(logP) + 0.6021(pI) + 1.1284(2)
where the multiple correlation coefficient R was 0.9837, the determination coefficient R^2^ was 0.9677, and the computed F-statistic was 29.9540. The IC_50_ values for the prepared compounds, for which it was not possible to determine them experimentally due to their poor solubility or a precipitate formation, or those that have not yet been synthesized, were estimated by the calculation from Equation (2). They are listed in Table 3 below together with their calculated values of log P and pI.

The synthesis and testing of the above-mentioned unsensitized compounds, as well as other structurally related ones, which would serve as a test set for the confirmation of Equation (1), are planned as the continuation of this research.

### 2.3. Proliferation Inhibitory Effects Induced by Thiosemicarbazides of Basically Substituted Acetamidoacetophenone Compounds in Human Cancer Cell Lines

For the target compounds that effectively inhibited APN activity, the antiproliferative activity in the different human cancer cell lines THP-1, MCF-7, and DU-145 was evaluated as IC_50_ values (50% inhibition concentration). The results are shown in Table 4, Figure 2, and the Appendix A. All five tested compounds significantly decreased the proliferation of THP-1 and MCF-7 cell lines in a concentration-dependent manner. On the contrary, only compound 24-10-3 induced an antiproliferative effect in DU-145 cells as well.

### 2.4. The Most Active Compounds and SAR

#### 2.4.1. SAR in APN Inhibition

The compound **24-11-2**, which is 4-[2-(4-methyl piperazine-1-yl)acetamido]acetophenone thiosemicarbazone hydrochloride 34 with an IC_50_ of 13.3 µmol/L, was found to be the most active compound, the activity of which was determined, whereas **22-4-1**, 2-[(diethylamino)acetamido]acetophenone thiosemicarbazone with an IC_50_ of 3.4 µmol/L was the most active in the series of compounds with the activities predicted using Equation 2. The overall activity results suggest that there is no significant difference in the activity of semicarbazones and thiosemicarbazones. The terminal basic parts, which seem to improve the activity, are 4-methyl piperazine-1-yl, 4-benzyl piperazine-1-yl, piperidine-1-yl, azepane-1-yl, and pyrrolidine-1-yl in most bulkier substituents with rather greater basicity (except for pyrrolidine-1-yl). As far as the influence of the positional isomerism is concerned, that is, the position of a substituted 2-aminoacetamido substituent regarding a Schiff base-containing group at the benzene ring, the results suggest that *p*-substituted compounds are more active than *m-* and *o-*derivatives.

#### 2.4.2. APN Inhibition vs. Antiproliferative Activity

The five compounds with the best values of IC_50_ for APN (ranging between 13 and 23.5 μmol/L) underwent testing for inhibition of cell proliferation on the three different cell lines, which differ from one another in their levels of APN expression. All five compounds triggered a significant antiproliferative effect in the cell lines expressing APN, THP-1, and MCF-7, whereas in the cell line DU-145 with no APN expression, four out of these five compounds did not affect proliferation at all. The remaining compound also inhibited DU-145 cell proliferation but less than in APN-positive THP-1 or MCF-7 lines (compare Figure 2). These results could suggest that the antiproliferative activity is linked with APN inhibition although other mechanisms that can also participate in it.

## 3. Discussion

The antiproliferative effect was determined for five selected compounds with the most potent APN inhibitory activity. Proliferation inhibitory effect determination was performed in three human cancer lines that differed from each other in their levels of APN expression. Although APN expression has been proved in the human monocytic leukemia cell line THP-1 [24] and breast carcinoma cell line MCF-7 [25], no APN expression was found in the DU 145 cells [26]. Our tested compounds showed a different effect in the cell lines used for analysis. All tested substances induced a significant antiproliferative effect in the cell lines expressing APN, THP-1, and MCF-7, whereas in the cell line DU-145 with no APN expression, the compounds **24-11-2**, **34-6-1**, **24-2-3**, and **24-5-3** did not affect proliferation at all. Compound 24-10-3 also decreased DU-145 cells but to a lesser extent than in APN-positive THP-1 or MCF-7 cells. In the case of this substance, we can consider another additional mechanism of antiproliferative action, but based on the data obtained, nothing closer can be stated. Although the obtained IC_50_ values against the APN-positive cell lines are at the level of double-digit micromoles, the substances represent an interesting model structure for the development of potentially therapeutically useful APN inhibitors.

The APN inhibition activity results suggest that the complexation of Zn^2+^ in the catalytic site of APN could be the mechanism underlying the inhibitory activity of basically substituted aminoacetophenones and their semicarbazones and thiosemicarbazones. A significant difference between the median inhibitory activity of the ketones (median IC_50_ = 395.7 μmol/L) and thiosemicarbazones together with the semicarbazones (median IC_50_ = 44.1 μmol/L) in compounds in which the IC_50_ values were experimentally determined, further suggests that the mechanisms of the Zn^2+^complexation with ketones and the Schiff bases derived from them must be different. This assumption was taken into account in the construction of the QSAR dependence that led to Equation (2). Based on the structure of (T-4)-[2-[1-[5-Acetyl-2-(hydroxy-κO)-4-hydroxyphenyl]ethylidene]hydrazinecarbothioamidato(2-)-κN,κN2]aquazinc [20] (Figure 3a), we proposed a possible mode of the complexation of Zn^2+^ cation with our basic thiosemicarbazones and semicarbazones. A tentative Zn^2+^ complex of **24-8-1** is an example of such a coordination compound, as seen in Figure 3b.

The 2^+^ charge of such a complex cation must be compensated by the negatively charged carboxylates belonging to the dicarboxylic amino acid residues, i.e., Asp or Glu. Glu320, which together with His297 and His361 takes part in Zn^2+^ complexation in the APN protein, and Glu264 or Glu298, which are nearby in the tertiary structure of the protein [27], can assume this role. Furthermore, when the whole group of Schiff bases with their determined anti-APN activities is separated into two groups—compounds with only one basic nitrogen atom (majority) and those with two basic nitrogens (piperazine derivatives)—the group of piperazines is markedly more active (median IC_50_ = 22.3 μmol/L) than the rest (median IC_50_ = 106.7 μmol/L). This fact could be caused by the possibility of the second nitrogen forming an ionic pair with a free carboxy group of another nearby dicarboxylic amino acid and thus interacting with the protein with a greater affinity. The water molecule, the complexation of which is expected in this model, is also a ligand naturally coordinated to Zn^2^ of APN [27] and we suppose that it remains coordinated when the spatial structure of the active site of the enzyme is changed by the binding of our ligand. A comparison of Figure 3a,b also suggests that the activity would benefit from the introduction of a chelating group into the *o*-position and an extension of the linker chain between the carbonyl of the acetamide group and the basic nitrogen to facilitate the formation of a donor–acceptor bond from the nitrogen to the zinc cation. This is the inspiration for our further synthesis of better APN inhibitors.

## 4. Materials and Methods

### 4.1. Chemistry

#### 4.1.1. General Information

All chemicals were purchased from commercial suppliers (Sigma-Aldrich, Darmstadt, Germany) and used as supplied without further purification. All reactions were monitored by TLC performed on precoated silica gel 60 F254 plates (Merck, Darmstadt, Germany). For compounds 12-2-1 to 14-14-1, ethyl acetate:hexane:diethylamine = 3:2:1 was used as an eluent, UV light (254 nm). For compounds **22-2-1** to **24-14-1**, petroleum ether: diethylamine = 9:1 was used as eluent, UV light (254 nm), and iodine was used for the detection of spots. NMR spectra were recorded on an FT-NMR ECZR 400 (JEOL, Akishima, Tokyo, Japan) spectrometer using TMS as an internal standard. The FTIR spectra were obtained with a Smart MIRacle™ Nicolet™ Impact 410 FTIR Spectrometer (Thermo Scientific, West Palm. Beach, FL, USA) equipped with the ATR ZnSe module. MS spectra were measured on a Xevo TQ-S triple quadrupole MS spectrometer (Waters, Milford, MA, USA) and analyzed in the positive mode under the formation of [M-H]^+^ ions. Melting points (uncorrected) were recorded on Kofler’s block Büchi Labortechnik AG 535 (BUCHI Labortechnik AG, Flawil, Switzerland). Detailed spectral and other structural data of the prepared compounds can be found in the Appendix A.

#### 4.1.2. General Procedure for the Preparation of **12-1**, **13-1**, and **14-1**

The 2-, 3-, or 4-aminoacetophenone 6.76 g (0.05 mol) was dissolved in 30 mL of acetone. Thereafter, 10 g (0.094 mol) of Na_2_CO_3_ was added. Then, 7.18 mL (0.09 mol) of 2-chloroacetyl chloride was added to the reaction mixture dropwise. The reaction mixture was stirred for 4 h at room temperature. After the completion of the reaction, 50 mL of 2M, HCl was added to the reaction mixture. The mixture was cooled at 0–5 °C overnight. The precipitate was filtered off, washed with distilled water, and dried to a constant weight.

#### 4.1.3. General Procedure for the Preparation of **12-2-1** to **14-14-1**

The synthetized 2-, 3-, or 4-(chloroacetamido)acetophenone 0.847 g (0.004 mol) was dissolved in 30 mL of acetonitrile. Then, K_2_CO_3_ 1.1 g (0.008 mol) was added to the mixture. Thereafter, 0.0044 mol of appropriate secondary amine was added to the suspension dropwise. The reaction mixture was stirred and refluxed for 4 to 8 h according to the secondary amine used. After the completion of the reaction (monitored by TLC), the reaction mixture was cooled at room temperature, potassium carbonate was filtered off, and the solvent was evaporated under reduced pressure to obtain the crude product. Synthetized 2-, 3-, or 4-[2-(dialkylamino)acetamido]acetophenones were washed with cooled distilled water and a small amount of ethanol and identified.

#### 4.1.4. General Procedure for the Preparation of **22-2-1** to **24-14-1** [28]

2-, 3-, or 4-[2-(dialkylamino)acetamido]acetophenone (0.004 mol) and thiosemicarbazide (0.008 mol) were added to 5 mL of 30% ethanol. The reaction mixture was refluxed for 3 h. After the completion of the reaction (monitored by TLC), the reaction mixture was kept at 0–5 °C overnight. The formed precipitate was then filtered off, washed with cold distilled water, and dried to a constant weight. The final products were recrystallized from ethanol in case of need.

#### 4.1.5. General Procedure for the Preparation of **22-2-2** to **24-14-2** [29]

The thiosemicarbazide (0.01 mol) and the appropriate 2-, 3-, or 4-[2-(dialkylamino)acetamido]acetophenone (0.01 mol) were dissolved in 20 mL of methanol. The reaction mixture was stirred for 10 min at room temperature and then an equivalent amount of 35% hydrochloric acid was added. The reaction mixture was refluxed for 5 h. After the completion of the reaction, the reaction mixture was cooled at 0–5 °C overnight. The precipitate was filtered off and recrystallized from 96% ethanol.

#### 4.1.6. General Procedure for the Preparation of **22-2-3** to **24-14-3** [30]

Thiosemicarbazide (0.01 mol) was added to 15 mL of distilled water and the reaction mixture was stirred and heated at 70 °C for 50 min to dissolve the thiosemicarbazide. Then, 2 mL of 50% perchloric acid was added to the mixture and the mixture was stirred for another 5 min at the same temperature. The solution of 2-, 3-, or 4-[2-(dialkylamino)acetamido]acetophenone (0.011 mol) in 2 mL of ethanol and 13 mL of distilled water was added to the reaction mixture. The mixture was heated and stirred at 85 °C for 3 h. After the completion of the reaction, the reaction mixture was cooled at 0–5 °C overnight. The precipitate was filtered off, washed with a small amount of cooled distilled water, and dried to a constant weight.

#### 4.1.7. General Procedure for the Preparation of **32-2-1** to **34-14-1** [31]

Semicarbazide hydrochloride (0.0025 mol) and sodium acetate (0.0025 mol) were dissolved in 10 mL of 96% ethanol. The mixture was stirred at room temperature for 15 min. Then, 0.0025 mol of synthesized 2-, 3-, or 4-[2-(dialkylamino)acetamido]acetophenone was added to the reaction mixture and the stirring continued at room temperature for the next 12 to 48 h. The reaction progress was monitored by TLC. The final product was filtered off, washed with ethanol, and dried to a constant weight.

### 4.2. Assessment of APN Inhibitory Activity

IC_50_ values against enzyme APN were determined using L-Leucine-*p*-nitroanilide (Sigma-Aldrich, Darmstadt, Germany) as the substrate and Microsomal Leucine aminopeptidase from porcine kidney EC 3.4.11.2, Type IV-S, ammonium sulfate suspension, 10–40 units/mg protein (Sigma-Aldrich, Darmstadt, Germany). The assay was performed by a Cytation 3 Cell Imaging Multi-Mode Reader (BioTek Instruments, Inc., Winooski, VT, USA) with appropriate Gen-5 software in 96-well plates. A 0.02 mol/L TRIS-HCl buffer solution at pH 7.5 was used as the assay buffer. All tested compounds were dissolved in TRIS-HCl buffer solution and compounds with poor solubility were dissolved in a small amount of DMSO or NMP used as a cosolvent. The assay mixture, which contained a variable amount of inhibitor solution (from 0 up to 80 μL), 10 μL of the enzyme solution, 5 μL of the substrate solution, and the assay buffer adjusted to 200 μL, was incubated at 37 °C for 40 min with short orbital shaking for 10 s. The hydrolysis of the substrate was monitored by the optical photometric method of absorbance in the visible and ultraviolet regions at a wavelength of 405 nm. The enzyme activity inhibitory rate was calculated from the measures of absorbance. The results of the 50% inhibitory activity of the enzyme (IC_50_) were determined through a regression analysis of the concentration/absorbance data by GraFit 5 software (Erithacus Software Ltd., East Grinstead, UK).

### 4.3. QSAR Statistic and Parameters Calculations

QCExpert 3.3 (TriloByte Statistical Software, Pardubice, Czech Republic) running onWindows 10 Education was used for the linear and multilinear regression calculations. ACD/ChemSketch was used for the log P values calculation by an algorithm based on >12,000 experimental logP values using the principle of isolating carbons [22]. MarvinSketch 6.2.2 (ChemAxon Ltd., Budapest, Hungary) was the software used for the isoelectric point pI values calculation by the algorithm built into it [23].

### 4.4. Evaluation of Proliferation Inhibitor Effects

#### 4.4.1. Reagents

All tested compounds were dissolved in dimethyl sulfoxide (DMSO) from Sigma Aldrich. Their fresh solutions were prepared prior to each experiment, whereas the final concentration of DMSO in the assays never exceeded 0.1% (*v*/*v*). RPMI 1640 culture medium, phosphate-buffered saline (PBS), fetal bovine serum (FBS), and antibiotics (penicillin and streptomycin) were purchased from HyClone Laboratories, Inc. (GE Healthcare, Logan, UT, USA).

#### 4.4.2. Cell Culture

A human monocytic leukemia cell line (THP-1), human breast adenocarcinoma cells (MCF-7), and human prostate cancer (DU-145) cell line were obtained from ATCC. THP-1 and DU-145 cells were cultivated in an RPMI 1640 culture medium and MCF-7 cells were cultivated in a DMEM medium, both supplemented with the antibiotic solution (100 U/mL of penicillin, 100 µg/mL of streptomycin) and 10% FBS. Cells were maintained in a humidified incubator with 5% CO_2_ at 37 °C and were regularly tested for the presence of mycoplasma contamination.

#### 4.4.3. WST-1 Analysis of Cell Proliferation

THP-1, MCF-7, and DU-145 cells were seeded in 96-well plates. Adherent cell lines were allowed to attach to the wells for 24 h. Cells were then treated with various concentrations of tested compounds to reach the final concentrations ranging between 1 µM and 100 µM and were incubated for 48 h. Cell proliferation was determined using Cell Proliferation Reagent WST-1 (2-(4-iodophenyl)-3-(4-nitrophenyl)-5-(2,4-disulfophenyl)-2H-tetrazolium) (Roche Diagnostics, Mannheim, Germany), as previously described [32,33]. WST-1 analysis was performed in three independent experiments, with each condition tested in triplicate. The IC_50_ values were determined using the nonlinear regression four-parameter logistic model using GraphPad Prism 5.00 software (GraphPad Software, San Diego, CA, USA). Statistical significance between the values was assessed by one-way analysis of variance (ANOVA) paired with Dunnett’s post hoc test using GraphPad Prism 5.00 software (GraphPad Software, San Diego, CA, USA) at levels of * *p* < 0.05, ** *p* < 0.01, and *** *p* < 0.001.

## 5. Conclusions

A series of 28 novel compounds based on the structure of 2-, 3-, or 4-[2-amino(acetamido)]acetophenone, where the amino group is a part of either a dialkylamino group or a saturated nitrogenous heterocycle, was synthesized. Twenty-two compounds from this series were tested for APN inhibitory activity. **24-11-2**, 4-[2-(4-methylpiperazine-1-yl)acetamido]acetophenone thiosemicarbazone hydrochloride with an IC_50_ of 13.3 µmol/L was the most active of them. A QSAR study with the semicarbazones and thiosemicarbazones revealed the relationship between the activity on lipophilicity expressed as logP and the acido-basic behavior of the compounds expressed as isoelectric point pI. Equation (1) describing this relationship enabled them to predict the activities of 33 other members of the semicarbazone and thiosemicarbazone series, 6 of which had already been prepared, and the remaining 27 were only proposed for synthesis. The compound with the best-calculated activity was **22-4-1**, 2-[(diethylamino)acetamido]acetophenone thiosemicarbazone with an IC_50_ of 3.4. The complexation of the Zn^2+^ cation of the active site of APN was proposed as the probable mechanism of activity, based on the similarity with other semicarbazones and thiosemicarbazones, which have served as ligands for the synthesis of transition metal complexes including those with Zn^2+^ as the central ion [16,17]. Five compounds with the best values of IC_50_ for APN (ranging between 13 and 23.5 μmol/L) underwent testing for inhibition of cell proliferation on three different cell lines that differ from each other in their levels of APN expression. All five compounds triggered a significant antiproliferative effect in the cell lines expressing APN, THP-1, and MCF-7, whereas in the cell line DU-145 with no APN expression, four of these five compounds, **24-11-2**, 4-[2-(4-methylpiperazine-1-yl)acetamido]acetophenone thiosemicarbazone hydrochloride, **34-6-1**, 4-[2-(piperidine-1-yl)acetamido]acetophenone semicarbazone, **24-2-3**, 4-[2-(diethylamino)acetamido]acetophenone thiosemicarbazone perchlorate, and **24-5-3**, 4-[2-(pyrrolidine-1-yl)acetamido]acetophenone thiosemicarbazone perchlorate, did not affect proliferation at all. The remaining compound **24-10-3**, 4-[2-(4-benzylpiperazine-1-yl)acetamido]acetophenone thiosemicarbazone perchlorate also inhibited DU-145 cell proliferation but less than in the APN-positive THP-1 or MCF-7 lines. These results suggest that the antiproliferative activity is linked with APN inhibition, although other mechanisms can also participate in it. Furthermore, our semicarbazones and thiosemicarbazones are the first compounds of these structural types of Schiff bases that were reported to inhibit not only a zinc-dependent aminopeptidase of the M1 family but also a metalloenzyme. The results, including Equation 2, can enable the proposal and synthesis of highly active APN inhibitors, which could serve as potential anticancer or antiviral drugs, which could contribute to overcoming the resistance of cancers to contemporary treatments.

## Data Availability

The data presented in this study are only available in this article and its Appendix A.

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
