# Peer review of "Aminopeptidase N Inhibitors as Pointers for Overcoming Antitumor Treatment Resistance"

_ijms, 2022, doi:10.3390/ijms23179813_

Round 1
Reviewer 1 Report (Previous Reviewer 1)

Author Response
Dear reviewer,
Thank you for your valuable comments that can help us to improve the quality of our manuscript. We attempted to perform most of your important recommendations. However, we cannot agree with all of them.
In Major concerns,
you criticized our system of numbering compounds and show us how to simplify it. However, our numbering system was developed in order to simply encode the structure within limited series of compounds, and also quickly identify the structure back from the code, including the intermediates. Our encoding, which we tried to construct as simple as possible, is explained in the legend in Scheme 1. It also can be expanded into further related series (another type of Schiff bases, further tertiary amines, other salts...). Well, we could change it into the numbering proposed by you, but the structure information is then lost and a table with the assignation of every compound to its number would be necessary for every look at compounds from the point of view of (qualitative) structure-activity relationships. Please, realize that eg. the structure, for which you proposed number 7, covers, in fact, three compounds that should have individual numbers: 2-[2-(dimethylamino)acetamido]acetophenone, 3-[2-(dimethylamino)acetamido]acetophenone, and 4-[2-(dimethylamino)acetamido]acetophenone. So, it would be necessary to characterize not only “R” as the basic residue, but also the position of benzene ring substitution, directly into the reaction scheme. The number of such labels would increase with the branching of the reaction scheme (thiosemicarbazones, carbazones, hydrochlorides, perchlorates…). Such a scheme would be, in my opinion, uncluttered. Furthermore, IJMS is not strictly an organic or medicinal chemistry journal, and its rules for authors do not prescribe any particular type of numbering.
The “twisted amino group”, in fact, the imino methylene group, should indicate that the configuration at this double bond is mixed E/Z, as we mention at the end of subchapter 2.1. This opinion was supported by NOESY (NMR) spectra measurements. The twisted bond symbol is in structure drawing editors, which we use, intended for the expression of such a situation – E/Z or trans/cis. We respect your recommendation and replace the twisted bond with a standard double bond and change the follow-up N-N single bond with the “undefined stereo bond”, which is a wavelet.
Subchapters 2.2. Determination of APN inhibition activity and 2.3. QSAR in APN inhibitory activity are now joined into one, 2.2 APN inhibitory activity and QSAR in it, as you requested.
We moved the paragraph about solubility problems and their solving to Supplementary Materials as you recommended. This paragraph was ordered on demand from one of the reviewers of the previous version of this manuscript.
You also criticized our NMR data presentation. Listings of all the spectra are in the Supplementary Materials and are complete. Our 13C-NMR, which we added also in graphical form only as examples, are not spectra with uncorrected phases, but APT (=attached proton test) spectra, in which carbons with an odd number of hydrogens are up of the baseline and those with an even number of hydrogens are down. (The solvent triplet at 77 ppm for CDCl3 goes also down.) Or the direction of peaks can also be a reversal (odds down, evens and solvent up), this depends on the result of phasing. Such a type of spectra is easier to interpret than classical 13C. (It is written under each such picture that it is apt).
In Minor concerns,
We respect your notice about the “sudden appearing” of M1 class at the end of the abstract and mention the membership of APN in the M1 family (which is better than M1 class and in better accordance with IUBMB recommendations) in the first sentence of the abstract now.
We also changed the order of paragraphs in our Discussion as you had proposed. You are right, the evaluation of the antiproliferative activity of compounds is more important than a debate about zinc chelation as a proposed mechanism of activity.
We appreciate all your comments, proposals, and criticisms. We know that many things between chemistry and biology remain that we need to learn.
Best regards,
Oldřich Farsa,
In the name of all authors

Reviewer 2 Report (New Reviewer)
The topic of article “Aminopeptidase N inhibitors as pointers on the way to overcome antitumor treatment resistance“submitted to me for review is very relevant and significant.
All methods and the methodologies are correctly and fully presented, and explained, and allow the reader to reproduce the experiments if needed.
The version of the manuscript I have to review is a corrected version of remarks already made. In this regard, after thoroughly reading the article, as well as thoroughly reviewing the supplementary material, I would like to give my positive remarks on the presented article.
I have no critical comments or recommendations.
I suggest this article be published as it is.
Author Response
Dear reviewer,
thank you very much for your helpful and encouraging opinion of our manuscript.
Best regards,
Oldřich Farsa,
in the name of all authors
Reviewer 3 Report (New Reviewer)
in attachment

Author Response
Dear reviewer,
Thank you for your valuable comments that can help us to improve the quality of our manuscript.
We appreciate the very professional, helpful, and matter of fact approach you took in evaluating our
manuscript.
We attempted to work all your recommendations and comments into the corrected version of
our manuscript. Our responses to your comments are as follows:
1. The purity of our compounds, expressed as HPLC chromatography homogeneity,
ranged between 91.6 and 99.9
2. The yields of our final compounds ranged typically between 45 and 70 %. We put
both types of data, yields and purity values, into a new table at the end of sub chapter
2.1 Synthesis of target compounds.
3. Compounds were purified by a simple crystallization f rom the system ethanol/water
with the addition of charcoal in the case of need. This was added to the text of sub
chapter 2.1.
4. QSAR in APN inhibitory activity:
a) The predicted, ie. computationally optimized activity data, calculated for
compounds, which w ere finally used for the construction of Equation 2, are not
significantly different from the experimental data. When the sets of experimental and
computationally optimized log IC 50 values are compared as two data samples, they
are not different in terms o f variance and mean, and distribution. There is a
significant correlation between these two data sets with the correlation coefficient
R(x,y) = 0.9145. We added columns with both experimental and optimized values
log IC 50 into Table 2. The predictivity of the model is satisfactory.
b) We think that values of
the multiple correlation coefficient R=0.9 837 and the
determination coefficient R 2 = 0.9677 for our final Equation 2, which was used for
estimation of activities of compounds, for which the experimental determination was
not possible, are satisfactorily high. It is necessary to note that this is a correlation
between the experimentally determined biological activity and 2 independent
calculated structure parameters. There are articles by renowned authors (“classics”)
from the field of QSAR in which such coefficients are a bit lower (eg . Hansch , C. et
al.: Bioorg. Med. Chem. 11 , 617 (2003), doi: 10.1016/S0968 0896(02)00326 7); here,
R 2 = 0.939. Some colleagues are sometimes satisfied with markedly lower val ues (eg.
Vávrová et. al., Bioorg. Med. Chem. 11 , 5381 (2003), doi:10.1016/j.bmc.2003.09.034);
here, there is R 2 =0.74 for an equation expressing the relationship between the
enhancement ratio and permeability coefficient in a series of transdermal permeation
enhancers.
We also tried to correlate other structure parameters, such as pK a , molar
volume, molecular surface area, molecular refraction, etc., and also values of log P
calculated by other algorithms, and also values of the (retent ion) capacity factor log
k´from RP HPLC acquired from the same system, in which we were checking the
purity of our compounds, as a lipophilicity parameter instead of log P, but we did not
reach a better correlation than Equation 2.
5. We corrected our chelate s structure by removal of the inappropriate (III) at the
terminal nitrogen and tried to make the tentative structure of the complex in Fig3b a
bit clearer by ignoring the previous attempt at spatial arrangement including a chair
conformation of morpholine. The bonds with arrows denote coordination bonds.
We also negligibly extended our Discussion as you had advised.
We appreciate all your stimulating comments. We believe that now our
article would be acceptable for publication.
Best regards,
Oldřich Farsa
In the name of all authors

Round 2
Reviewer 1 Report (Previous Reviewer 1)
Dear authors,
I appreciate your modifications and comments. The article looks better now. I apologize if my comments were taken as a bad review. It wasn't my intention. I keep thinking that the enumeration of the figures is a little confused but not incorrect.
best
This manuscript is a resubmission of an earlier submission. The following is a list of the peer review reports and author responses from that submission.
Round 1
Reviewer 1 Report
In their manuscript, Oldřich Farsa et al. synthesized a series of acetamidophenones, and semicarbazones, and thiosemicarbazones as Aminopeptidase N (APN) inhibitors with a possible application in cancer. Although interesting, the authors do not show any experiments in cancer cell lines to support their conclusions, reducing the article’s interest and not meeting expectations.
Furthermore, the authors have neglected too many things and details in the article. Description, schemes, and supporting information, among many other aspects of the manuscript, are incorrect or incomplete. The authors should carefully revise their manuscript.
Comments
- The title says “Aminopeptidase N inhibitors as potential weapons for overcoming cancer treatment resistance”. However, the results don’t match with the title. These inhibitors are neither ‘new weapons’ nor do they “overcome cancer resistance”. The title should be adjusted to the results.
- In the abstract, the authors said “This implies that APN inhibitors could be potent angiogenesis suppressors”, but no information about the angiogenesis has been explained to give context. The authors should better explain this to clarify the message.
- Scheme 1 is not well organized. It should be improved. There is much more space and the authors should modify that. In step two, there is a mistake. It lacked one carbon in the “chloroacetyl” group. The final compounds have twisted the double bond. Why? The authors should change the codes. The reader is unable to follow.
- Perhaps, the SARS studies could appear in a different order, explaining first the SARS studies and then the inhibitory activity.
- In the discussion, the authors said “The activity results suggest that the complexation of Zn2+ in the catalytic site of APN can be the mechanism of inhibitory activity of basically substituted aminoacetophenones and their semicarbazones and thiosemicarbazones” However, they have not performed any experiment to suggests this mechanism. They could change this explanation, maybe, based on previous literature.
- Where are the NMR analyses in the supporting info? These are not essential analyses, why? The authors should provide bi- and tridimensional analyses of all molecules described in the article.
Author Response
Reply to Review 1
Dear reviewer,
Thank you for your valuable comments, which helped us substantially improve our manuscript. Here, there are replies to your comments:
- we carefully checked and (believe that) removed all discrepancies among individual parts of the manuscript
- we accepted your recommendation concerning the title of our manuscript, and we softened the sound of the title into “Aminopeptidase N inhibitors as pointers on the way to overcome antitumor treatment resistance.”
- we reduced information about the participation of APN in the angiogenesis process to only mention supported by citation because it´s difficult to find anything unambiguous and convincing about it in the literature
- we changed the organization of Scheme 1 to better clarity and added missing carbon in the chloroacetyl group; twisted double bonds designate the mixture of E and Z geometry isomers
- we, however, insist on the order of appearance of activity results and (Q)SAR studies because (Q)SAR results are derived from the experimental ones
- our “statement” about activity mechanism mediated by Zn2 complexation is only suggestion derived from literature data, which are, however, poor as far as semicarbazone and thiosemicarbazone complexes are concerned (our references 19 and 20). Otherwise, zinc chelation is the most frequently referred APN inhibition mechanism, and we are afraid that the only reasonable and convincing (please eventually compare our reference 14). We plan to support this opinion by experimental preparation of zinc chelates of our compounds and by testing of inhibitory activity of our compounds against other zinc-dependent enzymes, not only aminopeptidases.
- spectral characterization of our compounds: due to the simplicity of their structures and the significant structural analogy among them, a conventional assignment of signals to appropriate hydrogens and carbons is satisfactory. We also used 2D (bidimensional) spectra (COSY, HMQC, HMBC, NOESY) for better interpretation of 1D 1H- and 13C-spectra, as usual. We added examples of such spectra in graphical form into Supporting Materials. 3D (tridimensional) spectra are not typical and unnecessary for structure characterization for our simple compounds.
We believe that our revised manuscript could now be better acceptable for you.
Sincerely yours,
Oldřich Farsa,
in the name of all authors
Reviewer 2 Report
The article describes the synthesis of a small library of novel aminopeptidase N inhibitors, their computed logP and pI values, their experimental IC50 values and a QSAR model. Using the model, calculated IC50 values are reported for a series of not yet prepared derivatives, or derivatives not subjected to pharmacological assay due to solubility problems.
The work is clearly and logically presented. The text is well structured and appropriately backed up with references. However, the present version of the manuscript is more suited to a letter than a full article.
The introduction is lacking details on known inhibitors, particularly their structure and activity. The presentation of the relevant research landscape would need further discussion to allow the assessment of the originality, novelty and context of the present work. Despite an original coding system used, the presentation of the synthetic work would benefit from a scheme or table representing the structures of the novel derivatives. Such scheme or table could be part of the supplementary information. Although the text refers to MS used for characterization of the compounds, MS results are not presented neither in the text, nor in the supporting information.
The rationale behind choosing logP and pI as parameters best correlating with the activity (and used for the regression model) is not presented in sufficient detail.
Solubility issues met with several compounds and methods to circumvent them are not further addressed in the text, as well as any potential correlation with the logP values. Comparison of IC50 values with that of known inhibitors would help to put in context the obtained results.
The rationale behind the design of the not yet prepared compounds listed in Table 2 is not discussed in sufficient detail. Given the relative simplicity of the synthetic pathway, it is not justified in the text why these compounds were not prepared and used for validating the QSAR model. The calculated IC50 values of Table 2 were not confronted with experimental assays that would be an important confirmation of the approach and support claims made in the conclusion on the design of novel potent inhibitors.
In the supplementary information, experimental details are provided for 23-2-1, 23-3-1 and 23-5-1, indicated as ’not yet synthetized’ in Table 2. This discrepancy should be cleared.
Advances over the present state of the art are not convincingly outlined in the article, the SAR and mechanism of action observations are limited in their scope.
For the discussion of the synthetic work it would be highly recommendable to include the respective NMR spectra in the supplementary material, as well as complement the characterization of the novel compounds with MS or HRMS and HPLC purity results.
Specific comments:
-line 14 – „we’re prepared” reword to „were prepared”
-line 46-48: the sentence „Vicinal cycloaliphatic…” needs rewording
-Scheme 1: the structures of 12-1, 13-1, 14-1 do not seem correct (i.e. NHC(O)Cl vs NHC(O)CH2Cl forming probably with chloroacetyl chloride). The names indicated in line 71 are also not corresponding to the structures of 12-1, 13-1, 14-1 on Scheme 1.
-line 76-77: „reaction with semicarbazide gave thiosemicarbazides” should be checked
-line 80: „in form of perchlorate or hydrochloride” reword to „in form of perchlorate or hydrochloride salts”
-line 93: „inhibitory expressed as IC50” reword to „inhibitory activity expressed as IC50”
-line 100: number 3-8-1 should be checked (vs the XX-X-X numbering used)
-Table 1: the IC50 value indicated for 24-11-3 in Table 1 is not in correspondence with the value indicated in the text (line 117)
-line 141: the sentence „Zn2+ complex of..” needs rewording
-line 143: Figure 2 should be corrected to Figure 1
-Figure 2 would benefit from a 3D representation or a representation of the target site as well
-line 159: the sentence „All the reactions were monitored…” appears twice in the text, one should be deleted
-line 182: reword „to a secondary amine” to „to the secondary amine”
-line 183: reword „cooled at” to „cooled to”
Author Response
Reply to Review 2
Dear reviewer,
Thank you for your valuable comments, which helped us substantially improve our manuscript. Here, there are replies to your comments:
- we prolonged the manuscript's text so that now it perhaps better fits the category “article.”
- we added details about important APN inhibitors
- as far as originality and novelty are concerned, we newly emphasized that our Schiff bases are the first thiosemicarbazones and semicarbazones in which inhibitory activity against a metalloenzyme was demonstrated
- we added an overview table of our target compounds with their structures and activities
- we added values from MS spectra
- we tried to explain in more detail why logP and pI were chosen as structure parameters for the construction of our simple QSAR model
- we added a paragraph about solubility issues complicating the determination of the inhibitory activity of our compounds and methods for their overcoming, which we tried.
- aromatic semicarbazones and thiosemicarbazones are unique compounds that tend to be neither hydrophilic nor lipophilic. Solubility of them is limited in any media. This complicated activity determination and their synthesis and mainly purification. For example, it´s difficult to separate a thiosemicarbazone from the residues of starting thiosemicarbazide, and flash chromatography also fails. This is probably the main reason why we didn’t prepare enough compounds suitable for testing, which could be separated into a “training” set intended for QSAR model building and a “test” set for the model validation as usual. We´ll continue preparing and testing Schiff bases and improving our regression model(s).
- we added examples of 1H- and 13C-apt and some 2D (COSY, HMQC, HMBC, NOESY) NMR spectra into the Supplementary Materials. We think adding all the spectra in graphical form is unnecessary as compounds are homologous and spectra thus similar. 2D spectra were only used for more sure assignment of signals in 1H and 13C-spectra.
- we corrected all errors which you wrote under “Specific comments.”
We hope that our corrected manuscript is now more suitable for publishing than previously and would be worthy of your recommendation.
Best regards,
Oldřich Farsa,
in the name of all authors
Round 2
Reviewer 1 Report
I would like to thank the authors for their prompt response to my revisions.
The authors have done most of the changes that I proposed, giving answers to some of my concerns. They have modified the title, abstract, schemes, and uploaded some of the 1D and 2D NMR spectra, but not all, improving their manuscript.
The Aminopeptidases N inhibitors have been studied over more than 20 years, with some inhibitors being tested in pre-clinical and clinical trials (doi: 10.1111/j.1349-7006.2010.01826.x). The in vitro activity of these compounds didn’t show any improvement versus many other Aminopeptidase N inhibitors that have been published in the literature (doi:10.1016/j.biochi.2010.04.026). furthermore, they haven’t performed any experiments in cell lines that explain the effect observed in vitro. The authors should perform these experiments with the aim to demonstrate the potential effect of these drugs in cancer therapy to justify their claims.
Reviewer 2 Report
The discussion of the context of the work, the experiments and the results was supplemented by additional information and explanations. Specific issues previously raised were addressed in the revised version, namely: i) a table with the structures of the studied compounds was added to the supplementary information, ii) the characterisation of the compounds was supplemented by MS results, iii) Scheme 1 was corrected and made more reader-friendly. The different corrections and modifications made considerably improved the overall presentation of the work. Understanding the challenges faced upon the synthesis and purification of the novel compounds, some doubts remain however regarding the scope and volume of the research vs the scope of the journal and articles from related fields therein.
Specific comments:
-the IC50 value indicated for 24-11-3 in Table 1 is not in correspondence with the value indicated in the text (line 177)
-line 156: number 3-8-1 should be checked (vs the XX-X-X numbering used)